# Marine nitrogen fixation as a possible source of atmospheric water-soluble organic nitrogen aerosols in the subtropical North Pacific

Tsukasa Dobashi[1,2], Yuzo Miyazaki[2], Eri Tachibana[2], Kazutaka Takahashi[3], Sachiko Horii[3,4], Fuminori Hashihama[5], Saori Yasui-Tamura[5], Yoko Iwamoto[6], Shu-Kuan Wong[7,8], and Koji Hamasaki[7]

[1]Graduate School of Environmental Science, Hokkaido University, Sapporo, 060-0810, Japan
[2]Institute of Low Temperature Science, Hokkaido University, Sapporo, 060-0819, Japan
[3]Graduate School of Agricultural and Life Sciences, The University of Tokyo, Tokyo, 113-8657, Japan
[4]Now at Fisheries Resources Institute, Japan Fisheries Research and Education Agency, Nagasaki, 851-2213, Japan
[5]Department of Ocean Sciences, Tokyo University of Marine Science and Technology, Tokyo, 108-8477, Japan
[6]Graduate School of Integrated Sciences for Life, Hiroshima University, Hiroshima, 739-8521, Japan
[7]Atmosphere and Ocean Research Institute, The University of Tokyo, Kashiwa, 277-8564, Japan
[8]Now at National Institute of Polar Research, Tokyo, 190-8518, Japan

*Correspondence to*: Yuzo Miyazaki (yuzom@lowtem.hokudai.ac.jp)

**Abstract.** Water-soluble organic nitrogen (WSON) in marine atmospheric aerosols affect the water-solubility, acidity, and light-absorbing properties of aerosol particles, which are important parameters in assessing both the climate impact and the biogeochemical cycling of bioelements. Size-segregated aerosol and surface seawater (SSW) samples were simultaneously collected over the subtropical North Pacific to investigate the origin of WSON in the marine atmosphere. The fine-mode WSON concentration ($7.5\pm6.6$ ngN m$^{-3}$) at 200–240ºE along 23ºN defined as the eastern North Pacific (ENP) was significantly higher than that ($2.4\pm1.9$ ngN m$^{-3}$) at 135–200ºE, defined as the western North Pacific (WNP). Analysis of the stable carbon isotope ratio of water-soluble organic carbon (WSOC) ($\delta^{13}C_{WSOC}$) together with backward trajectory indicated that most of the observed WSON in the fine particles in the ENP originated from the ocean surface. We found positive relations among nitrogen fixation rate, dissolved organic nitrogen (DON) in SSW, and the WSON concentrations. The result suggests that reactive nitrogen (DON and ammonium), produced and exuded by nitrogen-fixing microorganisms in SSW, contributed to the formation of WSON aerosols. This study provides new insights into the role of ocean-derived reactive nitrogen aerosols associated with marine microbial activity.

## 1 Introduction

Ocean-derived atmospheric organic aerosols (OAs) play an essential role in cloud formation processes and subsequently impact radiation over the open ocean (Rosenfeld et al., 2019). Therefore, it is essential to understand the formation process, composition, and emission flux of marine aerosols to assess the ocean-to-cloud relationship and its impact on climate (Oreopoulos et al., 2008; Salter et al., 2008). In addition to primary emissions of OAs (POAs), the origin and formation processes of secondary OAs (SOAs) from the ocean surface remain unclear. Brüggemann et al. (2018) indicated that additional SOAs from the oxidation of photochemically active precursors, such as volatile organic compounds (VOCs) from the ocean surface, are important, especially in tropical/subtropical regions with low POA concentrations; such SOAs contribute up to 60% of additional OA mass. However, field measurements of marine SOAs have been limited.

Regarding the chemical components found in marine organic aerosols, water-soluble organic nitrogen (WSON) can affect the hygroscopicity, light-absorbing properties, and acidity of OAs as well as the global biogeochemical cycling of nitrogen (Mohr et al., 2013; van Pinxteren et al., 2017; Facchini et al., 2008; Nehir and Koçak, 2018). Miyazaki et al. (2011) demonstrated that marine organic aerosols are enriched in ON from marine biological origins in the western North Pacific. Altieri et al. (2016) also suggested the importance of a marine biogenic source of aerosol WSON in the western North Atlantic,

leading to the conclusion that only 27% of total nitrogen (TN) deposition to the global ocean is anthropogenic, which is significantly less than the previously estimated values of 48–80% (Duce et al., 2008; Kanakidou et al., 2012). The ocean has been recognized to act as a source to atmospheric ON (e.g., alkyl nitrates, aliphatic amines, and urea) and ammonia ($NH_3$); the chemical compositions of atmospheric ON include a wide variety of functional groups (Altieri et al., 2016). However, the

5 origin and formation process of WSON aerosols, particularly those of marine atmospheric aerosols remain unclear.

There is also substantial uncertainty in our understanding of the linkage between the formation of aerosol WSON and marine phytoplankton at the sea surface. Among the major groups of marine phytoplankton, cyanobacteria are the dominant primary producers in subtropical gyres (Rousseaux et al., 2012; Ottesen et al., 2014), which occupy one-third of Earth's surface. In particular, $N_2$-fixing microorganisms including cyanobacteria are widely distributed in the subtropical North Pacific

(Cheung et al., 2020; Zehr et al., 2020), which is a primarily oligotrophic area with a weak vertical supply of nitrate from below the euphotic layer (Shiozaki et al., 2017; Yamaguchi et al., 2021). In this oceanic region, $N_2$ fixation by marine microorganisms such as *Trichodesmium*, a genus of filamentous cyanobacteria, is expected to be a possible source of reactive nitrogen in the atmosphere (Frischkorn et al., 2018). However, the effects of $N_2$-fixing microorganisms on the sea-to-air emissions of reactive nitrogen species including WSON, have not yet been investigated.

The present study aimed to elucidate the origin and formation process of WSON aerosols in the oligotrophic subtropical ocean, particularly their linkage with $N_2$-fixing microorganisms in surface seawater. We measured WSON and other reactive nitrogen species in size-segregated aerosols collected onboard a research vessel sailing over the subtropical North Pacific along the longitudinal transect in summer 2017. The discussion is extended to comparisons of the observed nitrogen aerosols with $N_2$-fixation rates in SSW to investigate possible sources of WSON over the oceanic region.

## 2 Experimental

### 2.1 Aerosol Sampling

Ambient aerosol sampling was conducted from 12 August to 5 October 2017 onboard the research vessel *Hakuho Maru* Hashihama et al., 2020; Yamaguchi et al., 2021). Sampling was carried out during cruise KH–17–4 from Vancouver to Tokyo

via Honolulu in the subtropical North Pacific (**Fig. 1**). The aerosol samples were collected using a high-volume air sampler (HVAS; Model 120SL, Kimoto Electric, Osaka, Japan) located on the deck above the bridge of the ship. A cascade impactor (CI; Model TE-234, Tisch Environmental, Cleves, OH, USA) was attached to the HVAS to collect size-segregated particles at a flow rate of ~1130 L min$^{-1}$, without temperature or humidity control. In the present study, we used analytical results obtained from the one bottom stage and four upper stages of the impactor, which collected particles with aerodynamic diameter

($D_p$) < 0.95 μm every 12 or 24 h and $D_p$ > 0.95 μm every 96 h, respectively. Aerosol particles collected at the bottom and upper stages were referred to as fine and coarse particles, respectively. To avoid possible contamination from ship exhaust during aerosol sampling, the sampling pump of the HVAS was shut off when the relative wind direction was out of ±60 degrees to the bow and/or when the relative wind speed was low (<5 m s$^{-1}$).

The aerosol samples were collected on quartz fiber filters (25 cm × 20 cm), which were precombusted at 450 °C for 6 h to

35 remove any contaminants. Each collected filter was stored in glass jars with a Teflon-lined screwed cap at −20°C to limit chemical reactions on the filter and losses of volatile compounds. In this study, aerosol samples collected with a total air volume < 100 m$^3$ were not used. In total, 51 and 9 samples are presented for the fine and coarse particles, respectively. These numbers accounted for 88% and 82% of the total numbers of fine and coarse aerosol samples collected, respectively.

## 2.2 Surface seawater (SSW) sampling and chlorophyll (Chl) a analysis

SSW samples were collected overboard at the depth of 0 m using an acid-cleaned bucket during each aerosol sampling duration. The average temperature of SSW at each sampling point was $26.1 \pm 2.1°C$. In total, analytical results of 12 SSW samples obtained along 23°N were shown in the present study. The SSW samples for Chl a analysis were filtered using 25-mm Whatman GF/F filters (GE Healthcare, Buckinghamshire, UK). Chl a concentrations were further measured fluorometrically using 10-AU™ Field and Laboratory Fluorometer (Turner Designs, Sunnyvale, CA) after extraction with N′, N′-dimethylformamide (Suzuki and Ishimaru,1990).

## 2.3 Chemical analysis of water-soluble aerosols and elemental carbon

To measure the concentrations of WSOC and WSTN in the aerosol filter samples, a filter cut of $19.63 \text{ cm}^2$ was extracted with 15 mL of ultrapure water under ultrasonication and filtered using a disc filter (Millex-GV, 0.22 μm, Millipore, Billerica, MA, USA). The concentrations of WSOC and WSTN were determined using a total organic carbon (TOC) analyzer with a TN unit (Model TOC-L$_{CHP}$+TNM-L, SHIMADZU). Additionally, another cut of the filter ($7.07 \text{ cm}^2$) was extracted with 10 mL of ultrapure water to measure the concentrations of $Na^+$, $NH_4^+$, $NO_3^-$, $NO_2^-$, and MSA. The same syringe filter type as described above was used, before the extract was injected into an ion chromatograph (Model 761 compact IC; Metrohm). Here the WSON concentration was defined as the difference between the WSTN and inorganic nitrogen ($NH_4^+$, $NO_3^-$, $NO_2^-$) concentrations. As an anthropogenic tracer, mass concentrations of elemental carbon (EC) were measured using a Sunset lab carbon analyzer. A filter punch of $1.54 \text{ cm}^2$ was used for the analysis of EC.

## 2.4 Stable carbon isotopic characterization of water-soluble organic aerosols

To determine the $\delta^{13}C$ of WSOC ($\delta^{13}C_{WSOC}$) in the collected aerosols, a portion of the filter ($9.08 \text{ cm}^2$) was extracted with 20 mL of ultrapure water. The extracted samples were then concentrated via rotary evaporation, and 40 μL of each sample was transferred to be absorbed onto 15 mg of precombusted Chromosorb in a precleaned tin cup. The $^{13}C_{WSOC}$ was then measured using an elemental analyzer (Flash EA 1112)/Continuous Flow Carrier Gas System (ConFlo)-Isotope Ratio Mass Spectrometer (Delta V, Thermo Finnigan) to determine the $\delta^{13}C$ of WSOC ($\delta^{13}C_{WSOC}$) (Miyazaki et al., 2012). The $^{13}C$ data were reported relative to an established reference of carbon Vienna Pee Dee Belemnite (VPDB).

## 2.5 Estimation of nitrogen fixation rate

Primary production and $N_2$-fixation rates were determined using methods previously reported (Shiozaki et al., 2017), which followed Mohr et al. (2010) and Hama et al. (1983), respectively. The SSW samples were collected directly from an acid-cleaned bucket into acid-cleaned 4.5-L polycarbonate bottles. The SSW samples to estimate the initial $^{15}N$ and $^{13}C$ enrichment of POM were filtered immediately onto precombusted GF/F filters. For incubation $^{13}C$-labeled sodium bicarbonate (99 atom% $^{13}C$; Cambridge Isotope Laboratories, Inc.) was added to each duplicate bottle at a final concentration of 200 μmol $L^{-1}$. The 110 mL of $^{15}N_2$-enriched SSW, in which 1.1 ml of $^{15}N_2$ gas (>99 atom% $^{15}N$, Shoko Science) was dissolved using a Sterapore membrane unit (20M1500A: Mitsubishi Rayon Co., Ltd.), was added to each bottle. The samples were then incubated for 24 h using an on-deck incubator with surface SSW running continuously. The incubation was terminated by gentle vacuum filtration of the SSW samples through precombusted GF/F filters. The filters were then frozen at $-20°C$ until the post measurement on the ground. The filter samples were dried at 50°C in an oven and were exposed to hydrogen chloride (HCl) fumes for 2 h to remove inorganic carbon, followed by being dried at 50°C again. The N and C contents and their stable isotope ratios were then determined using a DELTA V advantage mass spectrometer (Thermo Electron, USA) connected to an elemental analyzer. $N_2$-fixation activity was regarded to be significant when the atom% of $^{15}N$ for each incubated bottle was higher than that for the initial sample by 0.00146 atom% (Montoya et al., 1996). Primary production and $N_2$-fixation rates were determined twice at each sampling station.

## 2.6 Determination of dissolved inorganic and organic nitrogen

Surface seawater samples for the determination of dissolved inorganic and organic nitrogen were collected at 10-m depth using a conductivity-temperature-depth (CTD) system (Sea-Bird Electronics) equipped with acid-cleaned Niskin-X bottles (General Oceanics) (Hashihama et al., 2020). Total dissolved nitrogen (TDN) was measured by a persulfate oxidation method with a gas-segmented continuous flow analyser (Yasui-Tamura et al., 2020). Nanomolar concentrations of $NH_4^+$ and $NO_3^- + NO_2^-$ in seawater were determined by using sensitive liquid waveguide spectrophotometry (Hashihama et al., 2009; Hashihama et al., 2015). Dissolved organic nitrogen (DON) concentration was determined as the difference between the TDN and dissolved inorganic nitrogen (DIN = $NH_4^+ + NO_3^- + NO_2^-$) concentrations in surface seawater.

## 3 Longitudinal Distributions of WSON

**Figs.2a** and **2b** show the longitudinal distribution of the concentrations of WSON in fine particles ($WSON_F$) with particle diameter ($D_P$) < 0.95 μm and in coarse particles ($WSON_C$) with $D_P$ > 0.95 μm from samples collected at 23ºN in the subtropical North Pacific (**Fig. 1**). The average concentrations of $WSON_F$ and $WSON_C$ were 6.1±6.2 ngN m$^{-3}$ and 22±21 ngN m$^{-3}$, respectively, during the cruise measurement. The sum of these values is within a range of the WSON concentrations in total suspended particles (TSP) collected over the western North Pacific (3.0–35 ngN m$^{-3}$) (Duce et al., 2008). Here, along the cruise track at 23ºN, the oceanic region at 200–240ºE is defined as the eastern North Pacific (ENP), whereas the region at 135–200ºE is defined as the western North Pacific (WNP). Concentrations of WSON both in the fine and coarse particles showed a distinct longitudinal gradient, with substantially higher concentrations in ENP. The average concentrations of $WSON_F$ and $WSON_C$ in ENP were approximately 3–6 times as large as those in the WNP (**Table 1**).

The longitudinal distributions of the mass concentrations of inorganic nitrogen ($NH_4^+$, $NO_3^-$, and $NO_2^-$) generally showed similar patterns to those of WSON (**Fig. S1**). The sum of the mass concentrations of the inorganic nitrogen in ENP was approximately twice as large as those in WNP. Overall, $NH_4^+$-N was the dominant component of water-soluble total nitrogen (WSTN) (~78%) in the fine-mode aerosols, followed by WSON (16%) (**Fig. 2c**). In contrast, $NO_3^-$-N was the most abundant WSTN component of the coarse-mode aerosols, which accounted for 83% of the WSTN mass on average. Notably, the mass fraction of WSON in ENP was significant, accounting for up to ~50% (average: 19±15%) and ~11% of the WSTN in the fine and coarse particle masses, respectively (**Figs. 2c** and **2d**). This result indicates the importance of the abundance of WSON, particularly in the fine particles in ENP.

**Fig. 2e** presents the concentrations of chlorophyll (Chl) *a* in surface seawater (SSW) samples obtained during the cruise, as well as the average concentrations of Chl *a* during August–September 2017 derived from the MODIS-Aqua ocean color data (https://neo.sci.gsfc.nasa.gov/view.php?datasetId=MY1DMM_CHLORA&year=2017). The two different measurements of Chl. *a* generally showed similar longitudinal distributions, which resemble the longitudinal trend of the WSON concentrations. Moreover, 5-day back trajectories, calculated by HYSPLIT (https://www.ready.noaa.gov/HYSPLIT_traj.php), showed that the sampled air masses were transported primarily over the oceanic regions in the Pacific (**Fig. 1**). These results suggest that most of the observed aerosols were transported within the marine boundary layer with less influence from terrestrial sources prior to sampling. This is supported by the concentrations of nss-$Ca^{2+}$ and nss-$K^+$ as well as nss-$SO_4^{2-}$/methanesulfonic acid (MSA) ratios which did not show significant differences between WNP and ENP (**Table 1**). The overall results suggested that the observed aerosols, particularly those in ENP, were largely influenced by marine sources associated with phytoplankton.

# 4 Isotopic Characterization of Aerosol Organic Carbon and Formation Processes of WSON

Previously, a method using stable carbon isotope ratios ($\delta^{13}$C) of aerosol organic carbon has been successfully used to determine the contributions of marine and terrestrial sources to organic aerosols in the marine atmosphere (Cachier et al., 1986; Miyazaki et al., 2016). The $\delta^{13}$C of water-soluble organic carbon (WSOC; $\delta^{13}C_{WSOC}$) as a function of the $WSON_F$ concentrations is shown in **Fig. 3**. The average $\delta^{13}C_{WSOC}$ observed in this study was $-22.8\pm1.7$‰, with 71% of the data (12 out of 17 data points) in the range of typical marine origin (between $-24$‰ and $-18$‰) (Miyazaki et al., 2016), while that of terrestrial origin is between $-27$‰ and $-25$‰ (Cachier et al., 1986; Dasari et al., 2019). This isotopic characterization supports that most of the organic carbon fraction of the observed aerosols were of marine origin.

To elucidate the possible formation processes of the observed WSON, we used several tracers of marine origins. **Figs. 4a** and **4b** show scatter plots between WSON and $Na^+$ concentrations. The concentrations of $Na^+$, used as a tracer of marine primary emissions, did not show any significant positive correlations with those of $WSON_F$ ($R^2 = 0.04$) or $WSON_C$ ($R^2 = 0.03$) in the study area. Specifically, the $R^2$ value between $WSON_F$ and $Na^+_F$ was 0.003 in ENP, while $WSON_F$ and $Na^+_F$ concentrations even showed a negative correlation ($R^2 = 0.49$) in WNP (**Fig. 4a**). The correlation cannot be statistically discussed for the coarse mode separately in ENP and WNP due to the limited number of data points. Nevertheless, the WSON concentration was higher in the ENP than in the WNP for the same $Na^+$ concentration level in the coarse particles (**Fig. 4b**), suggesting that WSON was more abundant relative to $Na^+$ in coarse particles in ENP. Moreover, concentrations of glucose, a molecular tracer of marine primary aerosols (Miyazaki et al., 2018), were below the lower detection limit for most of the samples during the cruise. These results suggest insignificant contributions of direct emissions from the sea surface to the observed WSON.

The above results also imply that secondary formation was likely the dominant process underlying the formation of WSON observed in this study. The process includes accommodation of secondary WSON onto sea salt particles. **Figs. 4c** and **4d** present scatter plots between WSON and MSA concentrations in each particle size category. MSA has been widely used as a tracer of marine SOA because it is an oxidation product of dimethyl sulfate (DMS). MSA is either produced by gas-phase MSA directly scavenged by aerosols or rapidly produced in the aqueous phase from scavenged dimethylsulfoxide (DMSO) and methanesulfinic acid (MSIA) (Zhu et al., 2006). The concentrations of WSON did not show any significant correlations with those of MSA in each size range regardless of the oceanic region. The insignificant correlation suggests that the origin of the observed WSON aerosols differed from that of DMS or that the formation pathways of WSON were different from the oxidation processes of DMS. It is also possible that dependence of temperature and/or OH levels in the subtropics on the relative yields of MSA to sulfate in the oxidation of DMS (e.g., Mungall et al., 2018) might also partly affect the insignificant correlations between WSON and MSA. In fact, no relations were found between sulfate, known as an oxidation product of DMS, and WSON concentrations ($R^2 < 0.02$) in fine particles (data not shown). On the other hand, sulfate and WSON concentrations in coarse particles showed some positive relationships, although the number of data is limited. This may reflect overlapping processes of biogenic sulfate and WSON on a longer timescale (i.e., several days of sampling of coarse particles) even though the exact origins are different.

# 5 Distributions of nitrogen fixation rate, dissolved organic nitrogen, and aerosol WSON

To further explore the origin and possible formation process of the observed WSON aerosols associated with phytoplankton, we focused on $N_2$ fixation in SSW as a possible source of atmospheric reactive nitrogen. $N_2$ fixation is the biological conversion of $N_2$ to $NH_4^+$ or dissolved ON (DON), which represents the main external source of bioavailable nitrogen in marine environments. A significant fraction of fixed $N_2$ can be directly released by $N_2$-fixing microorganisms as dissolved inorganic nitrogen ($NH_4^+$, $NO_3^-$, and $NO_2^-$) and DON in the ocean. Data obtained from Station ALOHA in the subtropical North Pacific has shown enrichment of nitrogen pools (i.e., $NH_4^+$, $NO_3^-$, DON) within *Trichodesmium* blooms (Karl et al., 1992; Letelier

and Karl, 1994). It has been shown that majority of the recently fixed $N_2$ is released directly as DON during the growth of *Trichodesmium*, primarily as dissolved free amino acids (Capone et al, 1994; Glibert and Bronk, 1994). Luo et al. (2014) estimated that on average, spatially integrated $N_2$ fixation over the Pacific accounted for ~50% of that in the global ocean, pointing to the importance of the Pacific in terms of this process.

In the subtropical North Pacific, previously measured nitrogen isotope ratios ($\delta^{15}N$) in particulate organic matter (POM) from seawater suggested that POM was significantly affected by nitrogen supplied from $N_2$-fixing microorganisms (Horii et al., 2018). Indeed, in the current study, the average $\delta^{15}N$ of POM in the SSW samples was $-0.3\pm1.0$ ‰ (data not shown), which is within the range of $\delta^{15}N$ values of diazotrophic cyanobacteria typically ranging from $-2$ ‰ to 0 ‰ (Horii et al., 2018). This indicates that the organic matter in SSW collected in the study region is derived from nitrogen supplied by $N_2$-fixing
microorganisms such as cyanobacteria.

    **Fig. 5** shows the latitudinal distributions of the $N_2$-fixation rate and concentrations of DON and TDN in the SSW samples compared with those of aerosol WSON concentrations in each size category. The average $N_2$-fixation rate in the SSW samples collected in ENP was $38.6\pm13.3$ ngN $L^{-1}$ $d^{-1}$, which was significantly higher than that in the WNP ($11.5\pm10.6$ ngN $L^{-1}$ $d^{-1}$). The observed range of the $N_2$-fixation rate in SSW is similar to those reported for the same oceanic region in previous studies
(Montoya et al., 2004; Bonnet et al., 2009; Yamaguchi et al., 2019). Moreover, the longitudinal gradient of the $N_2$-fixation rate is similar to that along 23ºN in the tropical Pacific, as shown by previous field measurements and model simulations (Luo et al., 2014; Dutheil et al., 2018; Wang et al., 2019; Shiozaki et al., 2018; Hashihama et al., 2020). The higher $N_2$-fixation rate in ENP compared to that in WNP could be attributed to the greater abundance of $N_2$-fixing microorganisms such as *Trichodesmium* and a symbiotic unicellular cyanobacterium (UCYN-A) measured by quantitative polymerase chain reaction
(qPCR) of *nifH*. In particular, the number of the *nifH* copy of UCYN-A in ENP was three orders of magnitude larger than that in WNP (data not shown).

    The longitudinal distributions of DON clearly showed that DON concentrations in SSW of ENP ($4.77\pm0.53$ μM) were larger than those of WNP ($4.03\pm0.47$ μM) (**Fig.5c**). Furthermore, $N_2$-fixation rate and DON concentrations in SSW showed a positive correlation with $R^2$ of 0.63, where DON concentration accounted for >98% of TDN concentration, which showed the
dominance of DON in TDN during the cruise. These results suggest that the majority of DON in SSW were released associated with the recently fixed $N_2$ during the growth of $N_2$-fixing microorganisms.

    The WSON concentrations in both fine and coarse particles showed positive correlations with the $N_2$-fixation rates in the SSW samples (**Figs. 5** and **6**), with $R^2$ values of 0.27 ($p < 0.05$) and 0.60 ($p < 0.05$) in the fine and coarse modes, respectively. Additionally, concentrations of WSON and DON were positively correlated ($R^2 = 0.36$, $p < 0.05$). Meanwhile, the $R^2$ values
for the WSON concentrations and primary productivity in SSW were 0.08 ($p = 0.24$) and 0.58 ($p < 0.05$) in the fine and coarse modes, respectively (**Fig. S2** and **S3**), which were lower than those for WSON and $N_2$-fixation rate. The positive relation between the WSON mass concentrations and $N_2$-fixation rate in SSW suggests that reactive nitrogen produced by $N_2$-fixing microorganisms in SSW significantly contributed to the formation of WSON aerosols.

**6 Discussion on nitrogen fixation as a possible source of WSON in marine aerosols**

Previous laboratory experiments also showed that DON and $NH_4^+$ are released in seawater through $N_2$ fixation by microorganisms known as diazotrophs (Wannicke et al., 2009; Berthelot et al., 2015). Here, we discuss mass-based N:C ratios in seawater and atmospheric aerosols. The average WSTN:WSOC ratio in all the size ranges of the observed aerosols was $1.70\pm0.94$, while those in ENP and WNP were $1.61\pm0.99$ and $1.87\pm0.82$, respectively. It is noted that there may be bias in the WSTN:WSOC ratios particularly in WNP, because the WSOC concentrations were below the detection limit in many samples
(**Table S1**). Yvon-Durocher et al. (2015) reported that the N:C ratios of marine algal assemblages over the subtropical Pacific ranged between 0.10 and 0.13, while Wannicke et al. (2009) obtained N:C ratios of *Trichodesmium* ($0.21\pm0.02$) through a

laboratory experiment. Furthermore, Berthelot et al. (2015) reported that the ratio of dissolved nitrogen (DN) to DOC released during $N_2$ fixation was 0.07±0.48. The currently observed WSTN:WSOC ratio in the aerosols was much higher than the N:C ratios of microbes and the DN:DOC ratios in seawater affected by $N_2$ fixation. The higher WSTN:WSOC ratios in aerosols relative to the typical DN:DOC ratios in seawater suggest that nitrogen-containing aerosols are preferentially produced relative to organic carbon in the atmosphere.

It is possible to argue that anthropogenic sources might contribute to the observed WSTN including WSON and $NH_4^+$ as well as WSTN:WSOC ratios in aerosols shown above. Concentrations of EC as an anthropogenic tracer were below the lower detection limit (~0.1 μgC m$^{-3}$) in all the aerosol samples, where they did not show any statistically significant differences between WNP and ENP (Figure 5). This result suggests that effects of anthropogenic sources on the observed aerosols in ENP were likely small. This is consistent with the stable carbon isotope analysis, which suggested that most of the observed aerosols were of marine origin, rather than terrestrial sources including anthropogenic origin. Although measurement data of gas species was not available in this study, it is unlikely that only gas-phase precursors were of anthropogenic origin to explain secondary formation of WSON in spite that most of the observed aerosols were originated from ocean surface.

Previous studies showed that oceanic regions at low latitudes, including the subtropics act as a source of $NH_3$ (the net flux of $NH_3$ is out of the ocean to the atmosphere) (Jickells et al., 2003; Johnson et al., 2008). Paulot et al. (2015) used two global ocean biogeochemical models to show evidence for a missing source of atmospheric ammonia ($NH_3$ and $NH_4^+$) over the equatorial Pacific that was attributable to photolysis of marine ON at the ocean surface or in the atmosphere. In fact, DON was dominant component of TDN (>98%) in surface seawater, the longitudinal distribution of which was similar to that of WSON in our study (Figure 5). Furthermore, the observed $NH_4^+$ concentration levels in our study (ave. 40-50 ng m$^{-3}$) agree well with those predicted by Paulot et al. (2015) in the same oceanic region. Indeed, $NH_4^+$ was the dominant component of the aerosol reactive nitrogen in the fine particles (**Fig.2c**) in this study, whereas the correlations between $NH_4^+$ concentrations and $N_2$-fixation rates were insignificant ($R^2 = 0.06$). This insignificant correlation is partially attributable to phase partitioning of ammonia into the gas phase in the subtropical region. A possible source of $NO_3^-$ is not necessarily anthropogenic in the equatorial Pacific, while contribution of oceanic source in the open ocean cannot be ruled out. Several recent field studies suggested oceanic source of aerosol $NO_3^-$ in the eastern equatorial Pacific with evidence of extremely low values of $\delta^{15}N$ of $NO_3^-$ (e.g., lower than −5‰) (Kamezaki et al., 2019; Carter et al. 2021). In this oceanic region, alkyl nitrates are suggested to contribute to nitrate production, where relatively high concentrations of alkyl nitrates have been observed over the equatorial Pacific (e.g., Blake et al., 2003). The low concentrations of EC and stable carbon isotope ratios in aerosols together with the higher concentrations of DON in ENP partly supported oceanic source of aerosol $NO_3^-$ in this oceanic region, although $\delta^{15}N$ of $NO_3^-$ was not available in this study.

The secondary formation processes of WSON include emissions of gas-phase ON from the ocean and/or marine VOCs reacting with $NH_3$ (Paulot et al., 2015; Altieri et al., 2021). Specifically, aliphatic amines by gas-to-particle conversion are one candidate for WSON species observed in this study, as those amines of marine origin were observed in gas and aerosol phases in tropical/subtropical open oceans (e.g., Miyazaki et al., 2010; van Pinxteren et al., 2019). Although the exact mechanism of the WSON formation is not apparent in this study, the current results of the shipboard measurements suggest that $N_2$ fixation in SSW could partly explain one of the missing sources of atmospheric WSON and ammonia indicated by previous modeling studies.

To summarize, the current result suggests that $N_2$-fixing microorganisms in the SSW likely contributed to the formation of aerosol WSON and possibly other reactive nitrogen species, such as $NH_3/NH_4^+$, in the oceanic region of this study. Further field studies are required to elucidate the effect of $N_2$ fixation in surface seawater on the emission of atmospheric reactive nitrogen in different oceanic regions. This includes, for example, simultaneous measurements of gas and particle phases of organic and inorganic nitrogen species together with measurements of biological and chemical parameters relevant to $N_2$-fixing microorganisms in surface seawater by shipboard observations. Additional laboratory studies are needed to evaluate the

factors controlling the atmospheric emissions of reactive nitrogen associated with $N_2$ fixation in surface sea water. Jiang et al. (2018) predicted that global ocean warming in the future would result in large increases in growth and $N_2$ fixation by *Trichodesmium*. Consequently, the formation process and the amount of atmospheric WSON associated with $N_2$ fixation in surface seawater are expected to change, which should be important from the viewpoints of the climate effect of marine atmospheric aerosols as well as the air-sea exchange of nitrogen.

## 7 Conclusions

This study investigated the origin and formation process of WSON aerosols in the oligotrophic subtropical North Pacific, in terms of their linkage with $N_2$-fixing microorganisms in SSW, based on the cruise measurements. The average concentration of WSON in fine-mode aerosols along 23ºN in the eastern North Pacific ($7.5\pm6.6$ ngN m$^{-3}$) was much higher than that in the western North Pacific ($2.4\pm1.9$ ngN m$^{-3}$) during the research cruise. The stable carbon isotope ratio of WSOC together with backward trajectories indicated that most of the observed WSON in the fine particles in the eastern North Pacific originated from the ocean surface. Relations of the concentrations of WSON with those of Na$^+$ and MSA imply that secondary formation, that differed from the oxidation processes of DMS, was likely the dominant process underlying the formation of WSON observed in this study. Instead, significant positive correlations were found among nitrogen fixation rate, DON concentrations in SSW, and the aerosol WSON concentrations. Meanwhile, the EC concentrations in all the samples were below the lower detection limit and did not show any statistically significant differences between WNP and ENP, suggesting that effects of anthropogenic sources on the observed aerosols were likely small in this study. The overall results suggest that reactive nitrogen, such as dissolved organic nitrogen and ammonium, produced and exuded by nitrogen-fixing microorganisms in SSW, likely contributed to the formation of WSON aerosols over the oceanic region. This study provides new implications for the role of marine microbial activity in the formation of reactive nitrogen aerosols in the ocean surface.

**Data availability.**

All data used in this study are provided in the supplement.

**Author contributions.**

Y.M. designed and led the overall research. T.D. and Y.M. wrote the manuscript. T.D., E.T., and Y. M. performed the measurements of chemical parameters in the aerosol samples. K.T., S.H., Y.I., and K.H. collected the samples. S-K.W. and K.H. measured chlorophyll *a* concentration. S.H. and K. T. measured nitrogen fixation rate and $\delta^{15}$N of POM. F. H. measured nanomolar DIN and S. Y.-T. measured TDN in seawater. T.D., Y.M. and E.T. analyzed the data.

**Competing interests.**

The authors declare that they have no conflict of interest.

**Acknowledgements.**

We thank the researchers and crew of the R/V *Hakuho Maru* for their help with the observations. This research was supported by Grants-in-Aid for Scientific Research (16H02931, 19H04233, 24121005) from the Ministry of Education, Culture, Sports,

Science and Technology (MEXT), Japan. R. Yamaguchi is acknowledged for his help with the aerosol sampling. This study was also supported by the Grant for Joint Research Program of the Institute of Low Temperature Science, Hokkaido University.

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

**Table 1**: **Average concentrations of nitrogen species and stable carbon isotope ratios in the fine ($D_p < 0.95$ μm) and coarse ($D_p > 0.95$ μm) aerosol particles collected along 23ºN in each oceanic region.**

| | The western North Pacific (WNP, 135–200ºE) | | The eastern North Pacific (ENP, 200–240ºE) | | All (135–240ºE) | |
|---|---|---|---|---|---|---|
| | Fine | Coarse | Fine | Coarse | Fine | Coarse |
| WSON (ngN m$^{-3}$) | 2.4±1.9 | 7.9±6.0 | 7.5±6.6 | 49.0±12.6 | 6.1±6.2 | 21.6±21.3 |
| NH$_4^+$ (ngN m$^{-3}$) | 42.6±15.8 | 3.6±3.2 | 35.5±22.2 | 17.5±8.3 | 37.4±20.9 | 8.3±8.5 |
| NO$_3^-$ (ngN m$^{-3}$) | 0.5±0.4 | 154.6±65.8 | 1.8±1.8 | 373.2±91.1 | 1.5±1.7 | 227.5±127.6 |
| NO$_2^-$ (ngN m$^{-3}$) | 0.4±0.2 | 32.9±54.0 | 1.4±1.4 | 9.4±9.4 | 1.1±1.3 | 25.0±45.8 |
| nss-Ca$^{2+}$ (ng m$^{-3}$) | 1.6±1.5 | 166.4±81.5 | 0.4±1.1 | 211.8±4.3 | 0.7±1.3 | 181.5±69.9 |
| nss-K$^+$ (ng m$^{-3}$) | 5.2±4.8 | 120.5±60.7 | 3.9±3.0 | 141.8±9.9 | 4.2±3.6 | 127.6±50.9 |
| nss-SO$_4^{2-}$/MSA | 5.5±3.6 | 11.4±3.5 | 4.7±7.2 | 19.5±10.6 | 4.9±6.4 | 14.1±7.8 |
| $\delta^{13}C_{WSOC}$ (‰) | −23.9±1.4 | NA | −22.1±1.6 | NA | −22.8±1.7 | NA |

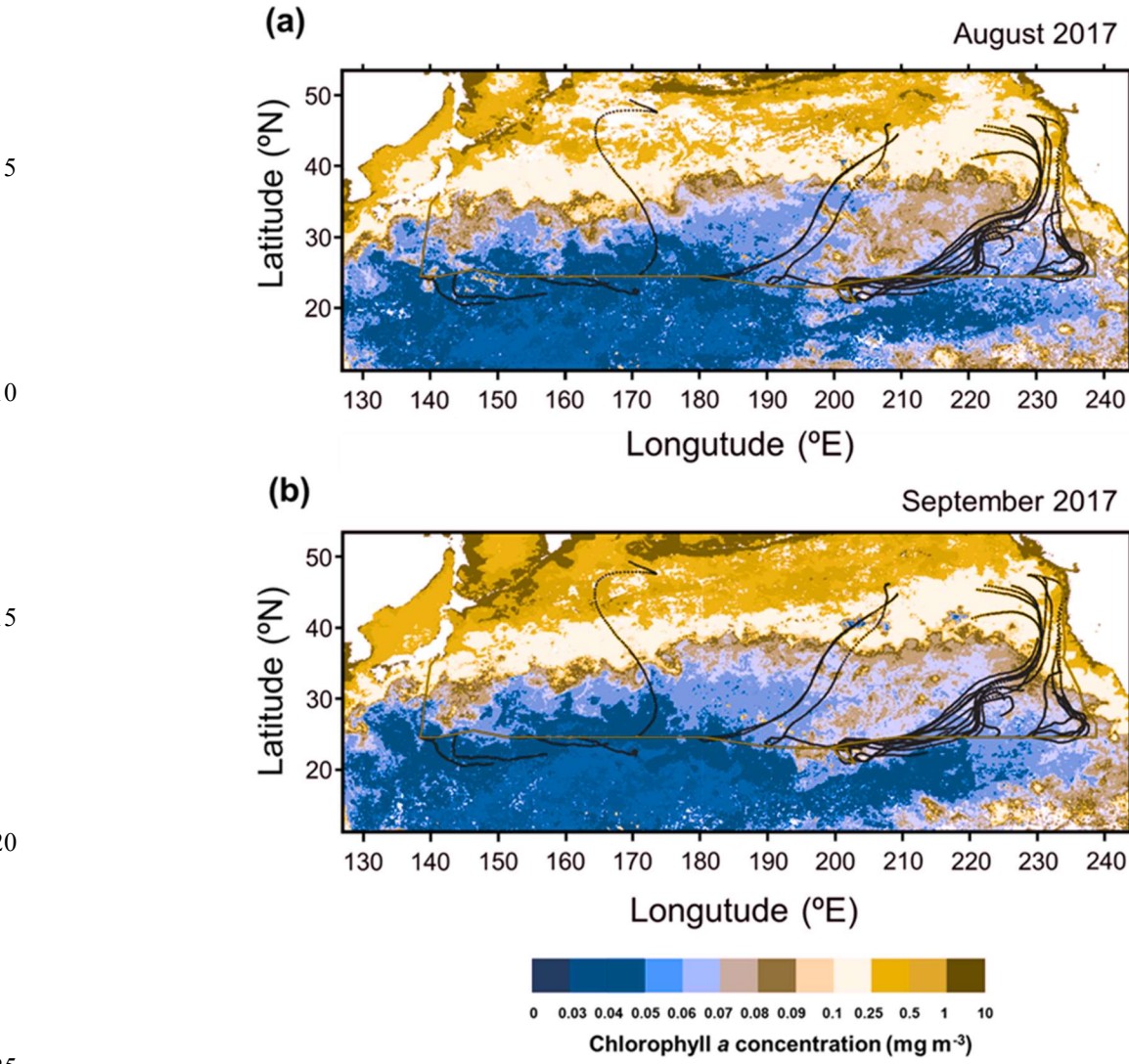

**Figure 1:** *R/V Hakuho* **cruise track in the subtropical North Pacific between 12 August and 5 October 2017 (red), together with typical 5-day back trajectories (black). Also shown are monthly averaged concentrations of chl a for (a) August and (b) September 2017 derived from the MODIS-Aqua (https://neo.sci.gsfc.nasa.gov/view.php?datasetId=MY1DMM_CHLORA&year=2017). Area in which data is missing is shown in white.**

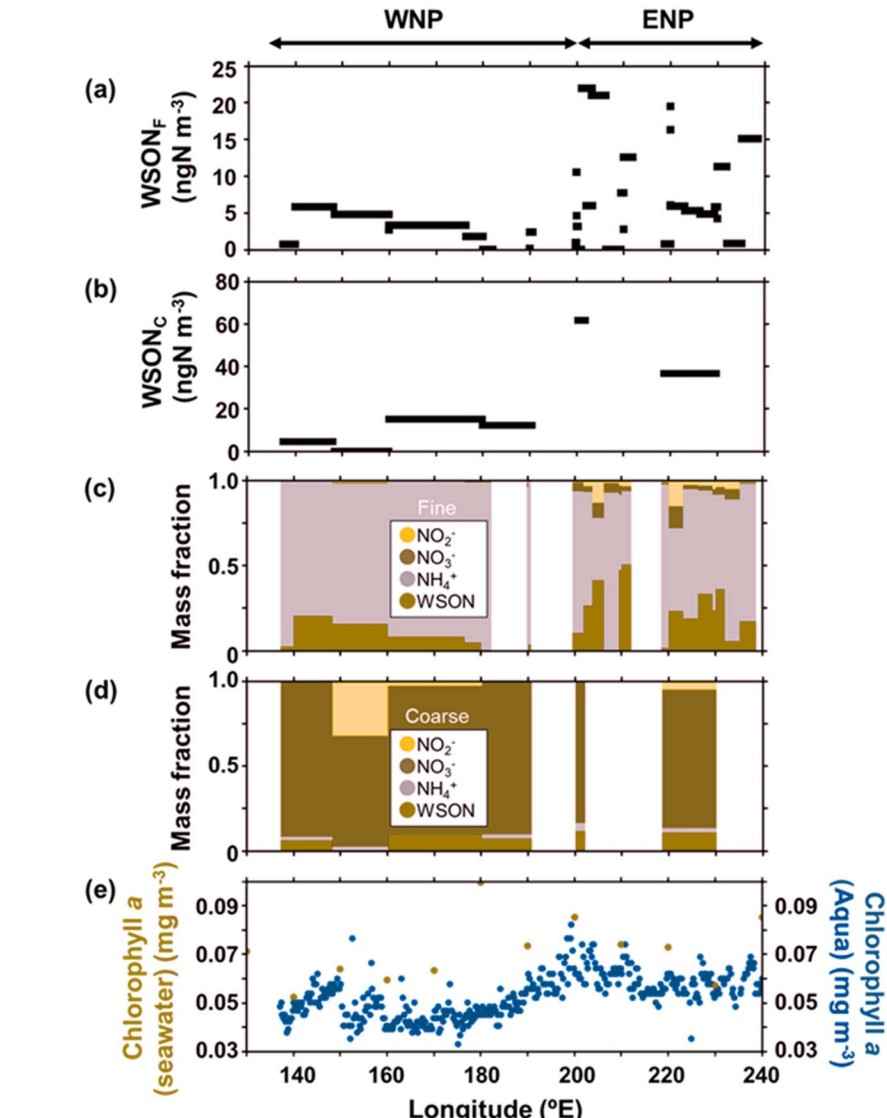

**Figure 2: Longitudinal distributions of each parameter of the atmospheric aerosols and surface seawater along 23ºN; the mass concentrations of WSON in (a) fine particles (WSON$_F$) and (b) coarse particles (WSON$_C$); the chemical mass fractions of nitrogen species in (c) WSON$_F$ and (d) WSON$_C$; (e) Chl *a* concentrations in the surface seawater samples during the cruise (ocher) and the average Chl *a* concentrations from August–September 2017, as measured by MODIS-Aqua (blue). WNP and ENP denote the oceanic regions of the western North Pacific and eastern North Pacific, respectively, defined in this study.**

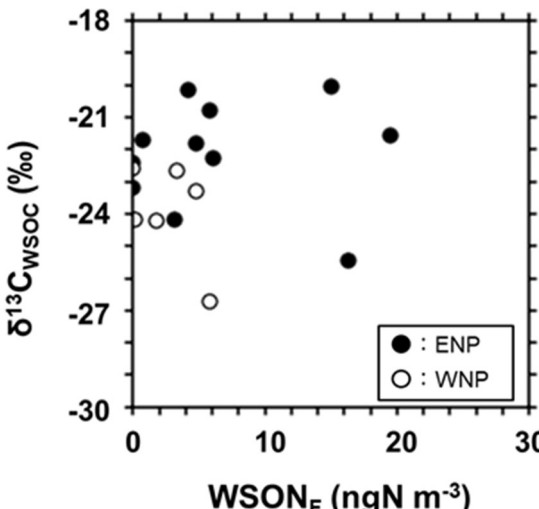

**Figure 3: δ¹³C of WSOC (δ¹³C$_{WSOC}$) as a function of the concentrations of WSON$_F$. Solid and open circles indicate the data of the eastern North Pacific (ENP; 200–240ºE) and the western North Pacific (WNP; 135–200ºE), respectively.**

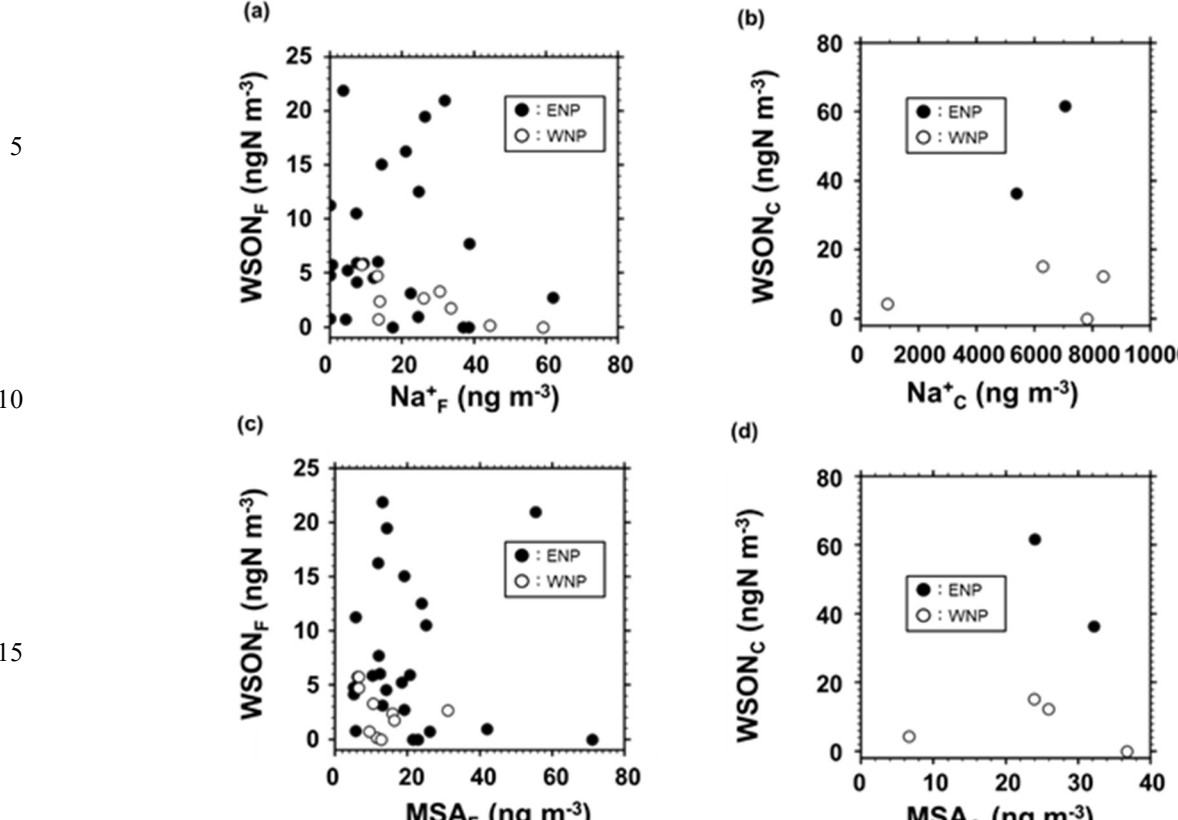

Figure 4: Concentrations of WSON as functions of those of sodium (a, b) and MSA (c, d), where (a) and (c) are for the fine particles and (b) and (d) are for the coarse particles. Solid and open circles represent the data of the eastern North Pacific (ENP) and western North Pacific (WNP), respectively.

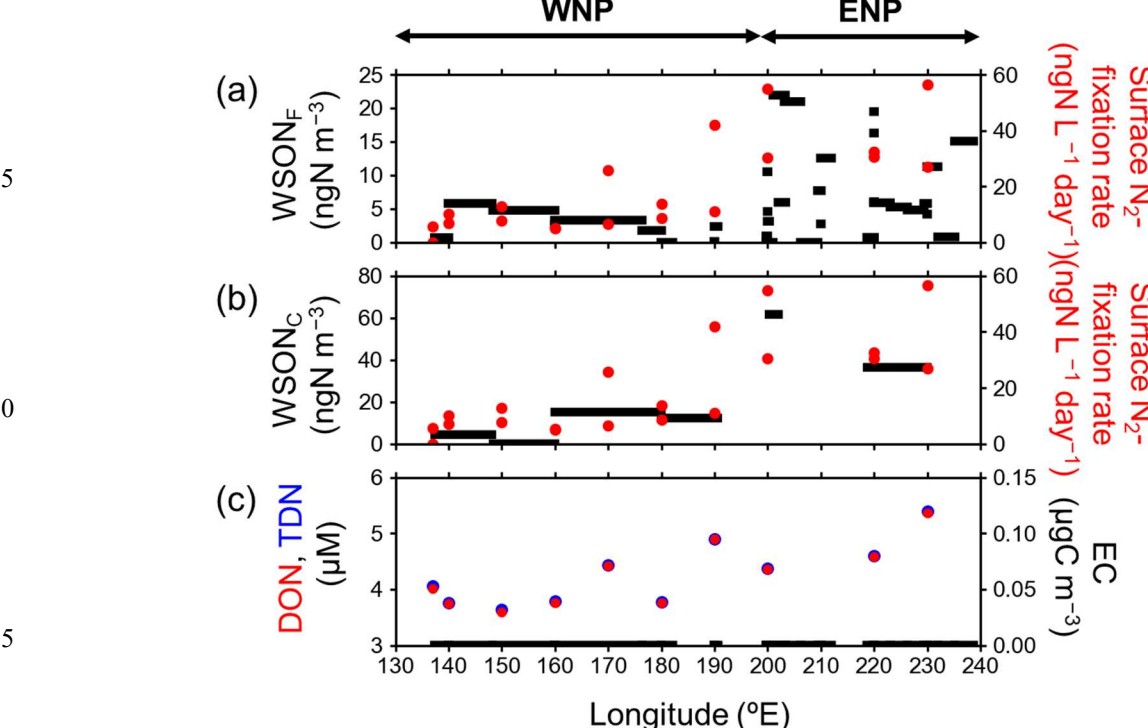

**Figure 5: Longitudinal distributions of mass concentrations of (a) WSON$_F$ (black) and (b) WSON$_C$ (black), together with N$_2$-fixation rate in the SSW samples (red solid circle), (c) DON (red solid circle) and TDN (blue solid circle) concentrations in surface seawater and aerosol EC concentrations (black) during the entire cruise.**

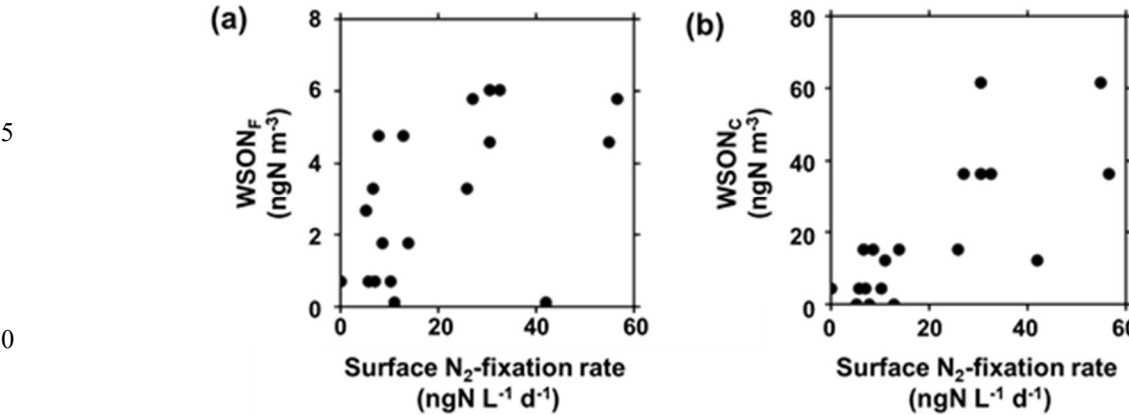

**Figure 6: Mass concentrations of (a) WSON$_F$ and (b) WSON$_C$ as a function of N$_2$-fixation rate in SSW. The data of N$_2$-fixation rate in SSW was merged into the duration of each corresponding aerosol sampling. For each one aerosol sample, one or more corresponding measurement data of N$_2$-fixation rate in SSW were obtained, so that the number of the data point in the panels is more than that of the aerosol samples.**