# Peer review of "Marine nitrogen fixation as a possible source of atmospheric watersoluble organic nitrogen aerosols in the subtropical North Pacific"

_EGUsphere, 2022_

## Author Comment (AC1)

**Responses to the comments of Referee #1:**

*Comment*: This paper attempts to address the uncertainty around sources of aerosol WSON in the marine atmosphere of the subtropical North Pacific. They report data from an east to west cruise transect across the N. Pacific including WSON aerosol concentration and surface ocean chl, primary productivity and N2 fixation rates. Their main approach is to compare east to west trends in the measured parameters. They conclude that since N2 fixation and aerosol WSON are both higher in the eastern N. Pacific than the western N. Pacific, then N2 fixation must be the source of aerosol WSON. The mechanism the authors invoke is that N2 fixation increases ammonium and DON concentrations in the surface ocean which then flux to the atmosphere and lead to secondary WSON. This paper is an example of "correlation does not equal causation." It is undeniable that the east to west trends are similar in N2 fixation and WSON aerosol concentrations, as in, they are both higher in the east than the west. But that could be due to multiple factors, and in no way suggests that one is causing the other. The authors proposed mechanism is completely untestable as they do not present ammonium or surface ocean DON concentrations. They suggest the WSON must be secondary as it does not correlate with sodium, but it also does not correlate with MSA, a classic indicator of secondary processing. I have chosen not to present a detailed review of the manuscript as the general framework presented is not supported in the literature, nor do the authors present a mechanism that can be tested by the existing data. The conclusions drawn are therefore based on a single correlation and are not supported in any way by what is presented in the paper.

*Reply*:  **We appreciate the referee's valuable comments on our work. Based on the comments, we added substantive discussion to construct compelling arguments by showing additional data as below.  Specific points of our major revisions are:**

**(1) To make a compelling argument on the causal relationship between $N_2$-fixation rate in surface seawater (SSW) and aerosol WSON concentrations, we have additionally shown and discussed the data of dissolved ON (DON) and total dissolved nitrogen (TDN) in SSW in the revised manuscript (Figure 5 has been revised as shown below). The figure clearly shows that DON concentrations in SSW of ENP (4.77±0.53 μM) were larger than those of WNP (4.03±0.47 μM). Furthermore, $N_2$-fixation rate and DON concentrations in SSW showed a positive correlation with $R^2$ of 0.63, where DON concentration accounted for >98% of TDN concentration, showing the dominance of DON in TDN during the cruise. The result suggests that DON were released associated with the recently fixed $N_2$ during the growth of $N_2$-fixing microorganisms. Based on these results, we have further discussed the linkage among $N_2$-fixation rate, DON, and aerosol WSON (P.6, L.22). This point is also discussed in terms of points below ((2) and (3)).**

**(2) Regarding (1), we have also added discussion on possible effects of anthropogenic sources on WSON by showing the data of elemental carbon (EC) concentration (Figure 5). The EC concentrations in all the samples were below the lower detection limit and did not show any statistically significant differences between WNP and ENP. This result suggests that possible effects of anthropogenic sources on the observed aerosols in ENP were small, which is consistent with the stable carbon isotope analysis in our study. We made additional statement on this point in the revised manuscript (P.7, L.6).**

**(3) In the revised manuscript, we also added literatures on dissolved nitrogen released by $N_2$-fixing microorganisms in the subtropical North Pacific, which support our discussion. Specifically, data obtained from Station ALOHA in the subtropical North Pacific has shown enrichment of N pools (i.e., $NH_4^+$, $NO_3^-$, DON) within *Trichodesmium* blooms (Karl et al., 1992; Letelier and Karl, 1994). It has been shown that majority of the recently fixed $N_2$ is released directly as DON during the growth of *Trichodesmium* (Capone et al, 1994; Glibert and Bronk, 1994). Indeed, our data showed that DON was a dominant component of TDN (>98%) in SSW, the longitudinal distribution of which was closely linked with that of $N_2$-fixation rate. In the revised manuscript, we made additional statement on these points (P.5, L.40).**

**(4) As we described in the original manuscript, the insignificant correlation between WSON and MSA suggests that the origin of observed WSON aerosols differed from that of DMS or that the formation pathways of WSON were different from the oxidation processes of DMS. We made additional discussion on this point with data of sulfate as below:**

**P.5, L.28:** *...It is also possible that dependence of temperature and/or OH levels in the subtropics on the relative yields of MSA to sulfate in the oxidation of DMS (e.g., Mungall et al., 2018) might also partly affect the insignificant correlations between WSON and MSA. In fact, no relations were found between sulfate, known as an oxidation product of DMS, and WSON concentrations ($R^2 < 0.02$) in fine particles (data not shown). On the other hand, sulfate and WSON concentrations in coarse particles showed some positive relationships, although the number of data is limited. This may reflect overlapping processes of biogenic sulfate and WSON on a longer timescale (i.e., several days of sampling of coarse particles) even though the exact origins are different.*

[Figure]

**Revised Figure 5: Longitudinal distributions of mass concentrations of (a) WSON_F (black) and (b) WSON_C (black), together with N$_2$-fixation rate in the SSW samples (red solid circle), (c) DON (red solid circle) and TDN (blue solid circle) concentrations in surface seawater and aerosol EC concentrations (black) during the entire cruise.**

**Another change:**

**·Along with the addition of DON and TDN data in the revised manuscript, Fuminori Hashihama and Saori Yasui-Tamura, who measured those parameters in seawater, have been added as co-authors.**

**References**:

Capone, D. G., Ferrier, M. D.,and Carpenter, E. J.: Cycling and release of glutamate and glutamine in colonies of the marine planktonic cyanobacterium, *Trichodesmium thiebautii*, Appl. Environ. Microbiol., 60, 3989– 3995, 1994.

Glibert, P. M. and Bronk, D. A.: Release of dissolved organicnitrogen by marine diazotrophic cyanobacteria, *Trichodesmium* spp. Appl. Environ. Microbiol., 60, 3996–4000, 1994.

Karl, D. M., Letelier, R., Hebel, D. V., Bird, D. F., and Winn, C. D.: *Trichodesmium* blooms and new nitrogen in the north Pacific gyre, p. 219-237, Marine pelagic cyanobacteria: *Trichodesmium* and other diazotrophs. Kluwer Academic Publishers, Dordrecht, The

Netherlands, 1992.

Letelier, R. M. and Karl, D. M: The role of *Trichodesmium* spp. In the productivity of the subtropical North Pacific Ocean, Mar. Ecol. Prog. Ser., 133, 263-273, 1994.

Mungall, E. L., Wong, J. P. S., Abbatt, P. D.: Heterogeneous oxidation of particulate methanesulfonic acid by the hydroxyl radical: kinetics and atmospheric implications, ACS Earth Space Chem. 2018, 2, 48–55, 2018.

---

## Author Comment (AC2)

**Responses to the comments of Referee #2:**

*Comment 1*: The paper by Dobashi et al. explores an important subject of secondary aerosol formation by nitrogen containing precursors over the open North Pacific. These types of studies are rare and difficult to come by and thus deserve to be published. The difficulty with the marine atmosphere research, especially in the Northern Hemisphere, is that it is often impacted by anthropogenic emissions via direct continental outflow or remnant terrestrial background. Multiple lines of evidence are, therefore, needed to rule out such impact and link atmospheric measurements to marine biological processes. The authors made a large effort in analyzing multiple tracers or species, but have not discussed the results in coherent and comprehensive manner. At the moment the paper is more like a measurement report without an in-depth analysis of underlying mechanisms or processes and often overselling correlation or similarity for a causal link. I encourage the authors to play devil's advocate and take an opposite side to assess if their argument can be falsified using same data. Therefore, a discussion where all evidence would be weighted pro and against is badly lacking. It may well be that the authors can compile compelling argument for their proposed link between nitrogen assisted secondary aerosol formation and nitrogen fixation in biologically active sea water, but a fair account should be given for the so called "null hypothesis" that observations were coincidental. It is also unclear how further studies should be geared in answering the remaining questions as a simple repetition will deliver same result.

*Reply 1*:  **We appreciate the referee's constructive and valuable comments. Taking fully account of the comments, we revised the manuscript to construct compelling arguments by showing additional data as below.  Our major revisions include the following points:**

**(1) Discussion on possible effects of anthropogenic sources on the observed aerosols has been added by showing the data of elemental carbon (EC) concentration, as suggested by the referee.**

**(2) To make a compelling argument on the causal relationship between $N_2$-fixation rate (and Chl. *a*) in surface seawater and WSON concentrations, we have additionally shown and discussed the data of dissolved ON (DON) and dissolved inorganic nitrogen (DIN) in surface seawater in the revised manuscript.**

**(3) We added substantive discussion on the points raised by the referee mainly in section 6.**

**Our responses to the specific comments and details of the changes made to the manuscript are given below.**

*Comment 2*: Abstract, line 22 "The result suggests..." (at best) and does not indicate, because the suggested link is only a proposition.

*Reply 2*:  **As the referee pointed out, the word "indicates" has been changed to "suggests" in the revised manuscript (P.1, L.23).**

*Comment 3*:  Page 2, line 31. The manual operation of sampling clearly helped to avoid own ship emissions, but what about transcontinental/continental outflow pollution? Air mass trajectories suggest no contact with land, but they typically do not rule out entrainment of pollution from decoupled boundary layer or free troposphere. Were there any tracers like black carbon measured to rule out anthropogenic impact and confirm marine origin of the sampled air mass? Note that higher WSON concentration observed over ENP where trajectories originated close to the continental boundary.

*Reply 3*: **Regarding possible influence of anthropogenic sources on the observed aerosols, the data of elemental carbon (EC) concentrations is additionally shown in Figure 5 as suggested by the referee. The EC concentrations in all the samples were below the lower detection limit (~0.1 µgC m$^{-3}$) and did not show any statistically significant differences between WNP and ENP. This result suggests that effects of anthropogenic sources on the observed aerosols in ENP were likely small, which is consistent with the stable carbon isotope ratios shown in our study. We made additional statement on this point in the revised manuscript (P.7, L.6).**

*Comment 4*: Page 2, line 37. Size segregated samples were collected every 48hours, but there were 51 fine and 9 coarse particles samples. More confusing are similar percentages of total samples collected. Please clarify why many coarse samples were missing in which case their percentage could not be as high.

*Reply 4*: **First of all, we apologize for erroneous description of the number of sampling duration. As we initially aimed to focus on fine particles, the filters for the fine particle were exchanged every 12 or 24 hours, while those for the coarse particles were exchanged much less frequently (~96 hours, not 48 hours). That is why the numbers of samples collected are different between fine and coarse particles. The percentages indicate the number fraction of samples toward the total number after the data screening based on the**

**total sampling volume for each. Therefore, the percentages should be similar for filters of fine and coarse particles. In the revised manuscript, we have corrected the corresponding numbers of sampling time (P.2, L.30).**

*Comment 5*: Page 3, line 23. Were SSW samples collected overboard with an acid cleaned bucket or using onboard plumbing system into acid cleaned bucket? In any case what was the approximate depth of SSW samples?

*Reply 5*: **SSW samples for the measurements of Chl.*a* and $N_2$-fixation rate were collected overboard with an acid cleaned bucket, where the depth of sampling was ~0 m. Meanwhile, seawater samples for DON (i.e., total DN (TDN) and DIN (Hashihama et al., 2009; 2015; Yasui-Tamura et al., 2020)), the data of which is newly added, were collected at 10-m depth using a conductivity-temperature-depth (CTD) system (Sea-Bird Electronics) equipped with acid-cleaned Niskin-X bottles (General Oceanics) (Hashihama et al., 2020). These descriptions have been added in the revised manuscript (P.3, L.2; P.4, L.2).**

*Comment 6*: Page 4, line 14. NH4 and NO3-N point to anthropogenic origin of nitrogen. NO3 in coarse particles results from chloride depletion by HNO3, while ammonia reduced ammonium ion is reacting with sulphate of whatever origin. Despite relatively low concentration of all inorganic species, their origin is pointing towards anthropogenic sources. It should also be noted that natural ammonia source over the oceans is almost negligible (unless the authors have evidence it is otherwise). I think that such discussion is essential, otherwise paper would look like a report instead of a scientific paper.

*Reply 6*: **Regarding possible origin of $NH_4^+$, Paulot et al. (2015) used global biogeochemical models with extensive ocean observational data to show that atmospheric NHx (= $NH_3$ + $NH_4^+$) fractions of oceanic origin were dominant (more than 70%) in the Equatorial Pacific (Figure 5 of Paulot et al. (2015)). The observed $NH_4^+$ concentration levels in our study (ave. 40-50 ng $m^{-3}$) agree well with those predicted by Paulot et al. in the same oceanic region (~20–60 ng $m^{-3}$; Figure 5 of Paulot et al.). Furthermore, Paulot et al. (2015) suggested that the photolysis of DON either in seawater or in the atmosphere may be a possible source of $NH_3$ in this oceanic region. In fact, DON was dominant component of TDN in surface seawater, the longitudinal distribution of which was similar to that of WSON in our study (see Reply7), which implies this oceanic source of $NH_3$.**

A possible source of $NO_3^-$ is not necessarily anthropogenic in the equatorial Pacific, while contribution of oceanic source in the open ocean cannot be ruled out. Several recent field studies suggested oceanic source of aerosol $NO_3^-$ in the eastern equatorial Pacific with evidence of extremely low values of $\delta^{15}N$ of $NO_3^-$ (e.g., lower than $-5‰$) (Kamezaki et al., 2019; Carter et al. 2021). In this oceanic region, alkyl nitrates are suggested to contribute to nitrate production, where relatively high concentrations of alkyl nitrates have been observed over the equatorial Pacific. Although we do not have measurement data of $\delta^{15}N$ of $NO_3^-$, we made additional discussions on the points of possible sources of $NH_4^+$ and $NO_3^-$ in the revised manuscript (P.7, L.17–30).

*Comment 7*: Page 4, line 22. Is the similar longitudinal distribution of chlorophyll coincidental or causal? Similarity or regression cannot prove causal relationship unless more substantial support is provided and thoroughly discussed. One way to look at it is to use some specific anthropogenic pollution tracers showing no connection or opposite relationship with WSON and inorganic species.

*Reply 7*: **To make convincing explanations on the causal relationship between Chl. *a* and WSON concentrations, we have added the data of DON and total DN (TDN) in surface seawater. As shown in the following Figure A, the longitudinal distribution of Chl.*a* concentrations was similar to those of $N_2$-fixation rate, DON concentrations in surface seawater, and aerosol WSON concentrations. In particular, $N_2$-fixation rate and DON concentrations in surface seawater showed a positive correlation with $R^2$ of 0.63, where DON was the dominant component (>98%) of TDN during the cruise. Furthermore, no relationship was found between these parameters and EC concentrations as an anthropogenic tracer, where EC concentrations in all the samples were below the lower detection limit with no significant difference in the longitudinal distributions (revised Figure 5 as shown below). Together with the stable carbon isotope ratios of aerosols, overall results suggest the causal relationship between $N_2$ fixation rate and WSON in this study. These descriptions were added in the revised manuscript (P.6, L.22). Furthermore, sentences regarding this point have been revised in the abstract (P.1, L.23) and conclusion section (P.8, L.15).**

[Figure]

**Figure A:** Longitudinal distributions of WSON$_F$ and DON concentrations in aerosols and suface seawater, respectively, together with those of nitrogen-fixation rate and chlorophyll *a* concentrations.

[Figure]

**Revised Figure 5**: Longitudinal distributions of mass concentrations of (a) WSON$_F$ (black) and (b) WSON$_C$ (black), together with N$_2$-fixation rate in the SSW samples (red solid circle), (c) DON (red solid circle) and TDN (blue solid circle) concentrations in surface seawater and aerosol EC concentrations (black) during the entire cruise.

*Comment 8*: Figure 3 is as useful as can be misleading as it contains only 17 of 51 fine aerosol samples and contains only 3 points with WSON >10ngN/m3. Most of the marine origin deltaC13 points (20-24 permille) probably belong to WNP region, so the highest value is of those 8 samples where WSON was >10ngN/m3. If those 8 points were eliminated, ENP and WNP regions become indistinguishable.

*Reply 8*:   **We do not intend to mention that ENP and WNP are different in terms of the isotopic characterization but intend to show that most of the aerosols were of marine origin observed during the entire cruise.**

*Comment 9*: Figure 4. Absence of the correlation between MSA and WSON could be due to the temperature impact which favours SO4 formation in subtropical latitudes instead of MSA. Have the authors looked at SO4 to WSON correlation. If the source of sulphur and nitrogen species is biogenic, they should correlate to some extent, because 48hour sampling has a footprint of ~1000-1500km and biogenic processes overlap even if underlying mechanisms differ somewhat.

*Reply 9*: **As the referee pointed out, dependence of temperature and/or OH levels in the subtropics on the relative yields of MSA to sulfate in the oxidation of DMS (e.g., Mungall et al., 2018) might partly result in the insignificant correlations between WSON and MSA. Scatter plots between sulfate and WSON are shown in Figure B below.  For the fine particle, no relation was found between sulfate and WSON concentrations ($R^2 < 0.02$). This supports that the origin of the observed WSON aerosols differed from that of DMS or that the formation pathways of WSON were different from the oxidation processes of DMS within a timescale of 12-24 hours. On the other hand, sulfate and WSON concentrations in coarse particles showed some positive relationships, although the number of data is limited. This may reflect overlapping processes of biogenic sulfate and WSON on a timescale of several days of coarse-mode aerosol samplings, as the referee pointed out. In the revised manuscript, we made additional descriptions on these points (P.5, L.28).**

[Figure]

**Figure B:** Scatter plots between WSON and sulfate in fine (left) and coarse (right) particles.

*Comment 10*: Page 5, line 11. The authors correctly discuss primary versus secondary origin of WSON species, but secondary formation does not necessarily is of marine origin. The real challenge is to rule out anthropogenic sources of gaseous precursors leading to secondary WSON.

**Reply 10: As described above, the EC concentrations in all the samples were below the lower detection limit, which did not show statistically significant differences between WNP and ENP. This suggests that contribution of anthropogenic sources on the observed aerosols in ENP were likely small, which is consistent with the stable carbon isotopic compositions in our study. Although we do not have measurement data of gas species, it is unlikely that only gas-phase precursors were of anthropogenic origin in spite that most of the observed aerosols were originated from ocean surface. Taking account of the referee's comment, additional statement on this point has been added to the text as follows.**

**P.7, L.6:** "*It is possible to argue that anthropogenic sources might contribute to the observed WSTN including WSON and $NH_4^+$ as well as WSTN:WSOC ratios in aerosols shown above. Concentrations of EC as an anthropogenic tracer were below the lower detection limit (~0.1 μgC $m^{-3}$) in all the aerosol samples, where they did not show any statistically significant differences between WNP and ENP (Figure 5). This result suggests that effects of anthropogenic sources on the observed aerosols in ENP were likely small. This is consistent with the stable carbon isotope analysis, which suggested that most of the observed aerosols were of marine origin, rather than terrestrial sources including anthropogenic origin. Although measurement data of gas species was not available in this study, it is unlikely that only gas-phase precursors were of anthropogenic origin in spite that most of the observed aerosols were originated from ocean surface.*"

*Comment 11*: Page 5, line 24. Why would N2 fixation result primarily in inorganic dissolved nitrogen and not organic like amines?

**Reply 11: In the original manuscript, we described that $N_2$ fixation results in not only $NH_4^+$ (inorganic DN) but also DON. Indeed, data obtained from Station ALOHA in the subtropical North Pacific has shown enrichment of N pools (i.e., $NH_4^+$, $NO_3^-$, DON) within *Trichodesmium* blooms (Karl et al., 1992; Letelier and Karl, 1994). It has been shown that majority of the recently fixed $N_2$ is released directly as DON during the growth of *Trichodesmium*, primarily as dissolved free amino acids (Capone et al, 1994; Glibert and**

**Bronk, 1994). In fact, our data showed that DON was a dominant component of TDN (>98%) in surface seawater. In the revised manuscript, we revised the sentence on these points (P.5, L.40).**

*Comment 12*: Page 6, line 18. Different ratio can arise from preferential production as well as from remnant anthropogenic source. Despite coming back to the same argument of anthropogenic origin I do not object authors argument about biogenic origin of WSON, but rather encourage the authors to look for more direct evidence in discounting anthropogenic origin.

*Reply 12*: **As we mentioned above, the EC concentrations in all the samples were below the lower detection limit and did not show statistically significant differences between WNP and ENP. Together with the results of stable carbon isotope analysis, the overall results support that contribution of anthropogenic sources to the observed WSTN:WSOC ratio was likely small. We made additional description on this point, the statement of which is shown in Reply 10 above.**

*Comment 13*: Page 6, line 27. What about amines as a WSON species?

*Reply 13*: **Aliphatic amines by gas-to-particle conversion are strong candidates for species of WSON observed in this study, as those amines of marine origin were observed in gas and aerosol phases in tropical/subtropical open oceans (e.g., Miyazaki et al., 2010; van Pinxteren et al., 2019). We have mentioned those amines in the revised manuscript as follows:**

**P.7, L.32:** *"... Specifically, aliphatic amines by gas-to-particle conversion are one candidate for WSON species observed in this study, as those amines of marine origin were observed in gas and aerosol phases in tropical/subtropical open oceans (e.g., Miyazaki et al., 2010; van Pinxteren et al., 2019). Although the exact mechanism of the WSON formation is not apparent in this study, the current results of the shipboard measurements suggest that $N_2$ fixation in SSW could partly explain one of the missing sources of atmospheric WSON and ammonia indicated by previous modeling studies."*

*Comment 14*: Page 6, line 40. Discussion section is recommended to discuss all information presented in previous chapters and discussing them simultaneously in a comprehensive manner. I

encourage the authors of playing devil's advocate for their own advantage in taking critical view to their own data.

*Reply 14*: **We added substantive discussion on the points above raised by the referee in section 6. We decided to add discussions within section 6, not in a new section, because some relevant statements have been already made in section 6 and it is more natural to add discussion in the same section to make a compelling argument. Along with the substantial addition of discussion, the title of section 6 has been changed to "Discussion on nitrogen fixation as a possible source of WSON in marine aerosols"**

**Others:**

**・With regard to the referee's comment on further studies that should be geared in answering the remaining questions, we made additional statement as follows:**

**P.7, L.42:** *"…Further field studies are required to elucidate the effect of $N_2$ fixation in surface seawater on the emission of atmospheric reactive nitrogen in different oceanic regions. This includes, for example, simultaneous measurements of gas and particle phases of organic and inorganic nitrogen species together with measurements of biological and chemical parameters relevant to $N_2$-fixing microorganisms in surface seawater by shipboard observations. Additional laboratory studies are needed to …"*

**・Along with the addition of DON and TDN data in the revised manuscript, Fuminori Hashihama and Saori Yasui-Tamura, who measured those parameters in seawater, have been added as co-authors.**

**References**:

Capone, D. G., Ferrier, M. D.,and Carpenter, E. J.: Cycling and release of glutamate and glutamine in colonies of the marine planktonic cyanobacterium, *Trichodesmium thiebautii*, Appl. Environ. Microbiol., 60, 3989– 3995, 1994.

Carter, T. S., Joyce, E. E., Hastings, M. G.: Quantifying Nitrate Formation Pathways in the Equatorial Pacific Atmosphere from the GEOTRACES Peru-Tahiti Transect, ACS Earth Space Chem. 2021, 5, 10, 2638–2651, 2021.

Glibert, P. M. and Bronk, D. A.: Release of dissolved organicnitrogen by marine diazotrophic cyanobacteria, *Trichodesmium* spp. Appl. Environ. Microbiol., 60, 3996–4000, 1994.

Hashihama, F., Furuya, K., Kitajima, S., Takeda, S., Takemura, T., and Kanda, J.: Macro-scale

exhaustion of surface phosphate by dinitrogen fixation in the western North Pacific, Geophys. Res. Lett., 36, L03610, doi:10.1029/2008GL036866, 2009.

Hashihama, F., Kanda, J., Tauchi, A., Kodama, T., Saito, H., and Furuya, K.: Liquid waveguide spectrophotometric measurement of nanomolar ammonium in seawater based on the indophenol reaction with o-phenylphenol (OPP), Talanta, 143, 374–380, 2015.

Kamezaki, K., Hattori, S., Iwamoto, Y., Ishino, S., Furutani, H., Miki, Y., Uematsu, M., Miura, K., and Yoshida, N.: Tracing the sources and formation pathways of atmospheric particulate nitrate over the Pacific Ocean using stable isotopes, Atmos. Environ., 209, 152–166, https://doi.org/10.1016/J.ATMOSENV.2019.04.026, 2019.

Karl, D. M., Letelier, R., Hebel, D. V., Bird, D. F., and Winn, C. D.: *Trichodesmium* blooms and new nitrogen in the north Pacific gyre, p. 219-237, Marine pelagic cyanobacteria: *Trichodesmium* and other diazotrophs. Kluwer Academic Publishers, Dordrecht, The Netherlands, 1992.

Letelier, R. M. and Karl, D. M: The role of *Trichodesmium* spp. In the productivity of the subtropical North Pacific Ocean, Mar. Ecol. Prog. Ser., 133, 263-273, 1994.

Miyazaki, Y., Kawamura, K., Jung, J., Furutani, H., and Uematsu, M.: Latitudinal distributions of organic nitrogen and organic carbon in marine aerosols over the western North Pacific, Atmos. Chem. Phys., 11, 3037–3049, https://doi.org/10.5194/acp-11-3037-2011, 2011.

Mungall, E. L., Wong, J. P. S., Abbatt, P. D.: Heterogeneous oxidation of particulate methanesulfonic acid by the hydroxyl radical: kinetics and atmospheric implications, ACS Earth Space Chem. 2018, 2, 48–55, 2018.

Paulot, F., Jacob, D. J., Johnson, M. T., Bell, T. G., Baker, A. R., Keene, W. C., Lima, I. D., Doney, S. C., and Stock, C. A.: Global oceanic emission of ammonia: Constraints from seawater and atmospheric observations, Global Biogeochem. Cycles, 29, 1165–1178, https://doi.org/10.1002/2015GB005106, 2015.

van Pinxteren, M., Barthel, S., Fomba, K. W., Müller, K., von Tümpling, W., and Herrmann, H.: The influence of environmental drivers on the enrichment of organic carbon in the sea surface microlayer and in submicron aerosol particles-Measurements from the Atlantic Ocean, Elementa Science of the Anthropocene, 5, 35, https://doi.org/10.1525/elementa.225, 2017.

Yasui-Tamura, S., Hashihama, F., Ogawa, H., Nishimura, T., and Kanda, J.: Automated simultaneous determination of total dissolved nitrogen and phosphorus in seawater by persulfate oxidation method, Talanta Open, 2, 100016,https://doi.org/10.1016/j.talo.2020.100016, 2020.

---

## Author Response (AR3)

**Our response to the comment of Referee #2:**

*Comment*: The authors made substantive and compelling argument in their revised manuscript to support their implied causal relationships in relation to WSON sources (and processes) of marine origin. The manuscript is now very significantly improved and deserves to be accepted for publication.

As for technical corrections I would like to note that "some positive correlation" presented in Figure B (response) should be presented with "p-value" which determines significance or insignificance of the correlation.

*Reply*: **We appreciate the referee's assessment and comment. According to the referee's technical comment, we added a p-value for the relation between sulfate and WSON concentrations in coarse particles. Although the p-value showed insignificance of the correlation, we believe that it does not affect our conclusions of the manuscript. The corresponding sentence has been revised as follows:**

**P.5, L.31;** *"···On the other hand, sulfate and WSON concentrations in coarse particles showed some positive relationships, although the number of data is limited and the relation is not statistically significant (p = 0.103)."*